# Impacts of Cipargamin on Na^+^-ATPase and Osmoregulation of *Trypanosoma cruzi*

**DOI:** 10.3390/membranes15120349

**Published:** 2025-11-23

**Authors:** Claudia Fernanda Dick, Ana Angelica Celso de Lima Barreto da Silva, Adriano de-Souza-Silva, Giovanna Frechiani, Juliana Barbosa-de-Barros, Adalberto Vieyra, José Roberto Meyer-Fernandes

**Affiliations:** 1Leopoldo de Meis Institute of Medical Biochemistry, Federal University of Rio de Janeiro, Rio de Janeiro 21590-902, Brazil; anaangelicaclbsilva@ufrj.br (A.A.C.d.L.B.d.S.); souza@biof.ufrj.br (A.d.-S.-S.); ju.bb998@biof.ufrj.br (J.B.-d.-B.); 2Graduate Program in Translational Biomedicine/BIOTRANS, Grande Rio University/UNIGRANRIO, Duque de Caxias 25071-202, Brazil; giovanna-degering@biof.ufrj.br (G.F.); avieyra@biof.ufrj.br (A.V.); 3Carlos Chagas Filho Institute of Biophysics, Federal University of Rio de Janeiro, Rio de Janeiro 21941-902, Brazil; 4National Center for Structural Biology and Bioimaging/CENABIO, Federal University of Rio de Janeiro, Rio de Janeiro 21941-902, Brazil

**Keywords:** *Trypanosoma cruzi*, P-type ATPase, cipargamin

## Abstract

The subfamily ENA of the P-ATPases family transports Na^+^ and H^+^ in opposite directions and, for this reason, they are called Na^+^-ATPases. They are physiologically important at high pH and high salt conditions in which other Na^+^ transporters cannot operate. This is the case, for example, of *Aspergilus* and *Arabidopsis*, respectively. Since, during their lifecycle, the parasitic protozoa face alkaline and/or high-saline environments, we postulated that the ENA subfamily could be a target for the treatment of serious and common illnesses driven by parasitic protozoa, as in the case of Chagas disease. The subgroup ATP4 ATPases, which is found in *Plasmodium falciparum* and *Toxoplasma gondii*, can be compared —through phylogenetic analysis—with the classic IID P-type ENA subfamily. Thus, some drugs that target PfATP4 ATPase and affect Na^+^ homeostasis are undergoing clinical trial, including spiroindolones. ENA-type ATPases (P-type ATPase IID) and ATP-type ATPase do not have structural homologs in mammals, appearing only in plants, fungi, and protozoan parasites, such as *Trypanosoma cruzi*, *Leishmania* sp., *T. gondii*, and *P. falciparum*. Therefore, this exclusivity points to Na^+^-ATPases as promising targets for medical projects aiming at new treatments contributing to the academic community.

## 1. Introduction

Chagas disease, caused by the protozoan *Trypanosoma cruzi*, is named after its discoverer [1]. Vertebrate hosts, for example, humans, can be infected through the infected feces of hematophagous triatomines through insect stings, skin fission, or mucous membranes (as conjunctiva, which causes the Romaña sign cf. [2]) or oral/digestive contact through contaminated food [3,4]. *T. cruzi* is found in diverse microenvironments, containing different solutes in varying concentrations. In most cases, life evolved in environments with high Na^+^ content [5], and these environments are considered inadequate or harmful to effective cellular functioning [6]. Thus, mechanisms that maintain low intracellular Na^+^ concentrations are necessary in unicellular organisms surrounded by an external fluid containing 150 mM Na^+^ [7]. Hence, the excretion of Na^+^ from the intracellular environment becomes difficult yet necessary for the survival, well-being, and even evolution of all organisms. Na^+^ flow across the plasma membrane was initially described as a concomitant K^+^ movement in the opposite direction, with the first Na^+^ pump described being the Na^+^/K^+^-ATPase [8], which is sensitive to ouabain. An ouabain-resistant Na^+^-ATPase activity with K^+^-independent action in the membranes isolated from guinea pigs’ external cortex was described later by Proverbio et al. [9]. Two mechanisms allow Na^+^ to pass through the basolateral membrane of proximal tubule cells and into the external environment: (i) one that is connected to K^+^ and sensitive to ouabain and (ii) another that is not coupled to K^+^ and is resistant to ouabain but sensitive to ethacrynic acid [10]. This ouabain-resistant K^+^-independent Na^+^ transport mechanism is called the “second Na^+^ pump” [11].

Na^+^-ATPases are part of the P-ATPase family, which forms a phosphorylated intermediate during the catalysis cycle, and they are involved in the transport of charged substrates such as Na^+^, K^+^, Ca^2+^, H^+^, Mg^2+^, Mn^2+^, Cu^+^, Zn^2+^, Cd^2+^, Pb^2+^, and phospholipids [11]. Furosemide inhibits Na^+^-ATPase without interfering with Na^+^/K^+^-ATPase [12], and this drug has significantly advanced Na^+^-ATPase identification in several organisms. The evolutionary advantage of Na^+^-ATPases is that they perform Na^+^ extrusion without interfering with intracellular K^+^ homeostasis. The Na^+^-ATPase of unicellular eukaryotes was first described in trypanosomatids [11], where ouabain, a specific inhibitor of Na^+^/K^+^-ATPase, did not completely inhibit the ATPase activity stimulated by Na^+^ in *T. cruzi* epimastigotes, suggesting the presence of a Na^+^-ATPase insensitive to ouabain in this parasite. The molecular identification of Na^+^-ATPase in *T. cruzi* was described by Iizumi et al. [13]. The gene responsible for Na^+^-ATPase in *T. cruzi* became known as TcENA because of its similarity to the gene for Na^+^-ATPase in plants and fungi, which is known as ENA (from the Latin *exitus natru*: exit of sodium). The TcENA present in *T. cruzi* is included in the Type IID P-ATPases group [14], which provides for ENAs from other microorganisms, such as *Leishmania braziliensis*, *L. donovani*, *Saccharomyces cerevisiae*, and *Entamoeba histolytica*. Analyzing the TcENA sequence, it is possible to identify 10 possible transmembrane domains (TMpred Server), a phosphatase motif, a site that is phosphorylated during the catalytic cycle, a nucleotide-binding domain, and residues involved in Na^+^ binding [11].

The Na^+^-ATPase of pathogenic protozoa has been considered essential to energize inorganic phosphate (Pi) fluxes through secondary active transport [15]. Furthermore, in apicomplexan parasites, it is evident that Na^+^-ATPase may be a therapeutic target for malaria and toxoplasmosis [14,16]. Regarding apicomplexans, the Na^+^-ATPase of these P-ATP4-type parasites does not have functional homologs to P-ATPases. Therefore, inhibition of these ATPases may impair parasite development and growth, and specific drugs against P-type ATPase could be designed and produced [16]. The group of synthetic compounds related to this has a promising pharmacological profile. It is a candidate for antimalarial drugs, because it can specifically inhibit the activity of P-type PfATP4 [17,18], and clinical trials are underway [19]. Dose-dependent concentrations of spiroindolones increase intracellular Na^+^ concentration, disrupting Na^+^ homeostasis and concomitantly inhibiting parasite proliferation. A very useful spiroindolone in humans infected with malaria is cipargamin (known as KAE609 or NITD609) [20]. Cipargamin disrupts parasites through oxidative damage, inhibits key metabolic pathways, and alters their structure within infected erythrocytes. These mechanisms occur together and promote the rapid and efficient elimination of parasites [21]. The *P. falciparum* mutant strain generated with cipargamin pressure produces a single mutated gene, PfATP4. Moreover, *pfatp4* mutants confer resistance to cipargamin, with a high fitness cost, indicating that cipargamin inhibits PfATP4 in a specific manner [22].

Na^+^ fluxes sustained by the ENA-type pump in trypanosomatids are linked to osmotic regulation, Na^+^ maintenance, and the activation of secondary active transporters, such as the Pi transporter [15]. Furthermore, inhibiting Na^+^-ATPase with furosemide in *Leishmania* interrupts cell proliferation in vitro. It reduces the lesion size and parasitemia in mice [23], demonstrating that this pump deserves to be explored in a biologically based project for new therapeutic interventions for parasitic diseases. The aim of this study is to evaluate the possible participation of Na^+^-ATPase in *T. cruzi* metabolism as a model for researching and developing new antiparasitic drugs to control emerging and reemerging diseases.

## 2. Materials and Methods

### 2.1. Cell Culture and Proliferation Profile

*Trypanosoma cruzi* epimastigote forms from the Dm28c strain (kindly supplied by Dr. Maria Auxiliadora Sousa, from Fundação Oswaldo Cruz/Fiocruz) were maintained in liver infusion tryptose (LIT) medium, supplemented with 10% fetal bovine serum (Cripion, São Paulo, Brazil) at 28 °C. In some experiments, low Pi LIT (with 2 mM Pi), or high Pi LIT (with 50 mM Pi) was used [15]. Sub-cultivation was performed every six days. For cell proliferation studies, the cell density was determined using a hemocytometer, and the proliferation curve was tracked from the same cell inoculum. Cell growth was monitored every day.

### 2.2. Cell Viability

Cell viability was assessed via a PrestoBlue Kit assay (Invitrogen, Carlsbad, CA, USA). Epimastigote cells subjected to cipargamin were washed with PBS buffer (pH 7.2). Cells were treated with 1% Triton X-100 for 30 min to provide dead cell-positive controls. After 10 min of incubation, the fluorescence was read (excitation 570 nm; emission 610 nm). The cell viability was expressed as a percentage relative to that of the untreated cells.

### 2.3. Na^+^-ATPase Activity in T. cruzi

The Na^+^-ATPase activity was measured as previously described [15], through Pi release from ATP hydrolysis. The parasites were collected by centrifugation at 1500× *g* for 15 min at 4 °C. They were then washed three times in a Tris-sucrose solution that included 20 mM KCl, 100 mM sucrose, and 50 mM Tris/HCl (pH 7.2). After this step, the cells were disrupted for 30 min in RIPA buffer, and Bradford’s assay was used to measure the protein concentration [24] method using BSA as a standard. Next, 100 µg of the homogenate was incubated for 60 min in a reaction mixture containing 20 mM HEPES-Tris (pH 7.0), 120 mM NaCl, 5 mM ATP, 10 mM MgCl_2_, and 1 mM ouabain, either with or without 1 mM furosemide or 50 nM cipargamin, as stated in the figure legend. The reaction was terminated by adding 1 volume of activated charcoal in 0.1 M HCl. The suspension was centrifuged at 1500× *g*, and an aliquot of the clear supernatant was added, in a 1:1 ratio, to the Fiske–Subbarow reagent (FeSO_4_, H_2_SO_4_, and (NH_4_)_2_MoO_4_). After 20 min of incubation, the results were determined spectrophotometrically at 650 nm.

### 2.4. Quantitative PCR

Following homogenizations of *T. cruzi* epimastigotes (1 × 10^8^ cells) in TRIzol (Invitrogen), RNA was extracted from the samples in accordance with manufacturer’s instructions. The Nanodrop ND-1000 (Thermo Fisher Scientific, Waltham, MA, USA) was used to measure the total RNA concentration spectrophotometrically. The High-Capacity cDNA Reverse Transcription Kit (Applied Biosystem, Waltham, MA, EUA) was utilized to create cDNA samples from five micrograms of total RNA, previously treated with RNase-free DNase I (Thermo Fisher Scientific, Waltham, MA, USA). TcENA (Na^+^-ATPase gene) expression was determined by qPCR analysis using the Applied Biosystems Step One Plus™ Real-Time PCR System. The parameters for using the SYBR Green PCR Master Mix (Applied Biosystems, Waltham, MA, USA) were as follows: a melt curve stage, 40 cycles of 15 s at 95 °C and 45 s at 60 °C, and one cycle of 10 min at 95 °C. The primers for TcENA were designed using the sequence obtained from the GenBank database (ID: AB107891). All experiments were performed in triplicate, and the expression of TcGAPDH (glyceraldehyde-3-phosphate dehydrogenase; GenBank ID: AF053742) was used for TcENA normalization. The primers used are described in Table 1.

### 2.5. Regulatory Volume Changes Under Conditions of Osmotic Stress

To measure osmotic stress under constant ionic strength, the following solutions were used: (i) Isotonic buffer containing: 64 mM NaCl, 4 mM KCl, 1.8 mM CaCl_2_, 0.53 mM MgCl_2_, 5.5 mM glucose, 5 mM Na-Hepes (pH 7.4), and 150 mM mannitol (320 mOsM); (ii) Hypertonic buffer, with same components as isotonic but with the mannitol concentration increased to 1.2 M (980 mOsM).

Epimastigotes grown for 6 days with or without 50 nM cipargamin were collected (1 × 10^8^ cells), washed with isotonic buffer at 37 °C, and resuspended in 100 µL of isotonic buffer. Then, the cells were transferred to a 96-well plate, and changes in the cell volume were measured by monitoring the absorbance at 550 nm in a plate reader. To prevent parasite decantation, the plate was read with continuous shaking. After monitoring the absorbance for three minutes, hyperosmotic stress was induced by adding 100 µL of hypertonic buffer to 100 µL of cells in isotonic buffer (final osmolarity, 650 mOsM). In control conditions, cells were submitted to 100 µL isotonic buffer. After inducing osmotic stress, the plate absorbance was measured for an additional 10 min at 550 nm. After 3 h of the cells under hyperosmotic stress, it was possible to assess a possible attempt of cell volume recovery. The cell viability was verified microscopically after 10 min of osmotic stress.

### 2.6. Immunofluorescence

Epimastigotes were incubated with 12 μM acridine orange for 30 min. Following washing in PBS, the cells were fixed for 1 h with freshly made 4% formaldehyde for 1 h. Following fixation, cells were attached to coverslips coated with poly-L-lysine and next permeabilized with 1% Triton X-100 for 10 min. A blocking solution containing 3% bovine serum albumin (BSA) diluted in PBS (pH 8.0) was used to incubate the samples. An α-tubulin antibody diluted in a blocking solution (1:1000) was then added to the slides and incubated for 1 h. Following a PBS wash, cells were incubated for 45 min with a 1:1000 dilution of Alexa488^TM^-conjugated anti-mouse IgG (Thermo Fisher Scientific, Waltham, MA, USA) in blocking solution. After 30 min of incubation with 5 μg/mL of DAPI, the cells were rinsed in a blocking solution. Using the anti-fading reagent ProLong Gold (Invitrogen, Waltham, MA, USA), slides were mounted and examined using a ZEISS LSM 910 confocal microscope (Carl Zeiss, Oberkochen, Germany).

### 2.7. Statistical Analysis

The average ± standard error (SE) from at least three separate experiments is represented by the values displayed in this study. GraphPad Prism 6 (GraphPad Software) was used to analyze these data using either Student’s *t*-test or one-way ANOVA, followed by Tukey’s multiple-comparison test. When the *p*-value was less than 0.05, differences between the data sets were considered significant.

## 3. Results

Epimastigotes forms of *T. cruzi* strain Dm28c exhibit Na^+^-ATPase activity, since they can hydrolyze ATP, stimulated by Mg^2+^ and Na^+^ ions, in the presence of the Na^+^/K^+^-ATPase inhibitor ouabain and independently of K^+^ (Figure 1A). For this reason, the following Na^+^-ATPase activity experiments are measured without K^+^ addition. It was also observed that furosemide inhibits this Na^+^-ATPase activity, supporting the hypothesis of a potential therapeutic target role for this enzyme (Figure 1B), with an IC_50_ for furosemide of 0.6 ± 0.1 mM.

Once furosemide significantly inhibited *T. cruzi* Na^+^-ATPase activity (Figure 1B), we tested the effect of furosemide on cell proliferation (Figure 2A). Furosemide (2 mM) significantly reduces the number of cells grown, compared to those cells grown without furosemide. Epimastigotes grown with 2 mM furosemide (5-day proliferation cultures) showed a significant decrease in Na^+^-ATPase activity compared to cells grown in the absence of furosemide (Figure 2B).

Furthermore, Na^+^-ATPase activity appears to be related to the response to nutritional stresses, such as the low availability of Pi. Low Pi in the culture medium increases the TcENA mRNA levels (Figure 3A) and Na^+^-ATPase activity (Figure 3B).

Then, cipargamin, a specific P-ATP4-type inhibitor [22], was tested on cell proliferation. Cipargamin efficiently disrupts cell proliferation at concentrations of 50 nM, compared to the control group (Figure 4A). In addition, cells maintained at 50 nM cipargamin present Na^+^-ATPase inhibition (Figure 4B), with a more accentuated effect than furosemide, whereas they present a similar inhibition with a concentration of the inhibitor that is 40,000 times lower (Figure 2B). A dose-dependent assay was performed to quantify the concentration dependence of the effect of cipargamin on Na^+^-ATPase activity (Figure 4C). Na^+^-ATPase inhibition was studied in the 1–30 μM cipargamin range.

Na^+^-ATPase inhibition induced by cipargamin may disrupt Na^+^ flux and influence the cell response to osmotic stress. To address this issue, cells grown in the absence or presence of 50 nM cipargamin were submitted to hyperosmotic stress (Figure 5A), and the light-scattering technique was used to measure their cell volume [25]. As observed, cells maintained in the presence of cipargamin shrank less in response to hyperosmotic stress than in control conditions. In addition, compared to the control, those cells cannot recover their cell volume after 3 h. With these data, it was possible to measure the regulatory volume increase (RVI) capacity using two different parameters [26]: the maximum cell volume change (Figure 5B) upon induction of hyperosmotic stress and the final volume recovery (Figure 5C) after 3 h. As observed, cells maintained in the presence of 50 nM cipargamin showed a significantly smaller change in cell volume (Figure 5B) and were less efficient in recovering their initial volume (Figure 5C) than the control group. This indicates an overall inefficient capacity to regulate cell volume in a hyperosmotic stress response, as observed in a significantly high ratio of cell recovery/final volume after stress (Figure 5D).

Acidocalcisomes are acidic organelles due to the presence of ion–proton translocators, which internalize large amounts of H^+^ into the lumen [27,28]. Changes in transporter activity and, consequently, in ionic composition can alter the acidic pH of acidocalcisomes [29]. Together with the contractile vacuole complex, acidocalcisomes play a fundamental role in pH regulation and osmoregulation [30]. To determine whether cipargamin affects the ionic composition and, consequently, the acid profile of acidocalcisomes, we used acridine orange, a fluorescent dye specifically designed for acidic compartments. Using immunofluorescence, we observed a change in the acid content compartmentalized in organelles, such as acidocalcisomes, responsible for cellular osmoregulation [31]. This acid content was measured by quantifying the fluorescence with acridine orange dye observed in cells maintained in the absence (CTRL) or presence (CIPA) of cipargamin. Clearly, cells maintained under cipargamin stress (CIPA) exhibit a decrease in this acid content, without changes in nucleic acid content or morphology (Figure 6).

## 4. Discussion

The main result of this study shows that Na^+^-ATPase plays an important role in *T. cruzi* metabolism, including Pi uptake and osmoregulation. Cipargamin, a Na^+^-ATPase inhibitor, can efficiently interfere with osmoregulation. These results illustrate the importance of parasite Na^+^-ATPases in cell proliferation and metabolism processes, especially in osmotic stress response. These Na^+^-ATPases were initially described as Ca^2+^-ATPases due to their structural similarity to the sarco-endoplasmic reticulum Ca^+^-ATPase (SERCA) [9]. Parasite Na^+^-ATPases (ENA-type or type IID P-ATPases) do not have structural homologs in mammalian cells, which makes them a promising drug target [11]. In terms of mechanisms, it should be emphasized that, in trypanosomatids, Na^+^ fluxes maintained by the ENA-type pump appear to be related to the maintenance of Δψ and the energization of secondary active transport, such as the Pi transporter in several parasites [15,32]. Furthermore, Na^+^-ATPase inhibition with furosemide stops cell *Leishmania* proliferation in vitro [11], demonstrating that this pump deserves more attention if it can be explored in the biologically based design of new therapeutic interventions for parasitic diseases. Similarly, Na^+^-ATPase inhibition with the classic inhibitor furosemide also disrupts *T. cruzi* epimastigotes’ proliferation (Figure 2). The role of Na^+^-ATPase in Na^+^ gradient maintenance, required for several essential secondary transporters, was also evaluated. In *T. rangeli* and *T. cruzi*, the Pi influx is linked to Na^+^-ATPase, without influence of (Na^+^/K^+^)-ATPase [15]. A Na^+^-dependent and a Na^+^-independent Pi uptake mechanism are present in both trypanosomatids. It is likely that, as a compensatory mechanism, when exposed to a low Pi medium environment, *T. cruzi* epimastigotes can upregulate Na^+^-ATPase activity and mRNA TcENA levels (Figure 3).

In apicomplexans, spiroindolones induce P-type ENA Na^+^-ATPase inhibition, with drugs targeting PfATP4 [33]. This inhibition causes a fatal disruption in Na^+^ homeostasis [18]. In silico analysis indicates that TcENA has a close structural similarity with PfATP4, and both proteins are close through phylogenetic analysis [11]. Cipargamin can also disrupt cell proliferation and Na^+^-ATPase activity in a dose-dependent manner in *T. cruzi* epimastigotes (Figure 4A,C). Parasites maintained under cipargamin stress present a downregulated Na^+^-ATPase activity (Figure 4B). This allows for the investigation of further effects regarding cipargamin inhibition. In this way, the inability of parasites maintained at cipargamin pressure to respond to hyperosmotic stress was evaluated (Figure 5). Osmolarity is one of the significant environmental obstacles that *T. cruzi* must overcome during its lifecycle in both vertebrate and invertebrate hosts [34]. *T. cruzi* presents a growth plasticity and metabolic flexibility, which is a determinant for parasite persistence through varied growth conditions or even under drug pressure [35]. For example, epimastigotes forms, found in the insect vector, face increases in osmolarity when passing through the insect digestive tract, with 1000 mOsm/kg in the rectal content [36]. In this way, pretreatment with cipargamin interferes with epimastigotes’ ability to shrink in response to hyperosmotic stress (Figure 5A,B), and these cells cannot recover their cell volume after 3 h. This recovery is known as regulatory volume increase (RVI), and it represents the ability of the cells to correct the rapid reduction induced by hyperosmotic stress. This RVI is mediated by ion transport systems, where the Na^+^/H^+^ exchanger plays a central role [37]. *T. cruzi* epimastigotes shrink in seconds when exposed to hyperosmotic stress, but they do not substantially restore their usual volume in the presence of cipargamin, indicating no fast uptake of water and inorganic ions [38].

As demonstrated previously, acidocalcisomes present different ion transporters on its membrane, as H^+^ (V-H^+^-ATPase, V-H^+^ pyrophosphatase), Ca^2+^ (Ca^2+^-ATPase), and other ions (Pi, Zn^2+^, Mn^2+^, Fe^2+^, polyamine, chaperone complex); and it also has exchangers (Na^+^/H^+^) through a variety of proteins [30,39]. The ability of *T. cruzi* to withstand the vast array of environmental circumstances it experiences throughout its lifecycle, including sharp variations in external osmolarity, is one unique aspect of its biology. As the insect’s feed status varies, these osmotic fluctuations also occur in the triatomine vector’s gut [37]. Poly P hydrolysis in acidocalcisomes have been shown to play a part in the regulatory volume drop that occurs in many trypanosomatids in response to hypoosmotic stress. So, alterations in poly P concentration could result in osmotic response defects [40]. For the synthesis of poly P to occur, the H^+^ gradient generated by V-H^+^-PPase is necessary [41]. Acidocalcisome alkalinization (Figure 6), detected as a fading in acridine orange staining, could be explained by a disruption in acidocalcisome ion fluxes. Na^+^/H^+^ exchanger plays a crucial role in response to hyperosmotic stress, being able to alkalinize this organelle in response to stress, whereas Na^+^/H^+^ exchanger inhibitor can disrupt the response to hyperosmotic stress [42,43]. It is already known that, in *T. cruzi*, activation of Na^+^/H^+^ exchanger in acidocalcisomes is favored by a gradient of Na^+^ between the extracellular media and the organellar lumen and induces calcium release from acidic vacuoles. Ca^2+^ release from acidocalcisome is crucial to respond to high osmolarity [43]. In this way, cipargamin treatment may disturb the Na^+^ gradient, interfering with Na^+^/H^+^ exchange in the acidocalcisome, and then the cells cannot respond to hyperosmotic stress. Further experiments may clarify this correlation.

## 5. Conclusions

In summary, it was observed that cipargamin inhibits TcENA in a dose-dependent way, leading to an intracellular Na^+^ accumulation. This Na^+^ excess should be accumulated in specialized organelles, such as acidocalcisomes. The Na^+^/H^+^ exchanger could mediate the Na^+^ entry; so, Na^+^ influx induces an acidocalcisome alkalinization. During hyperosmotic stress, there is an increase in protein hydrolysis to amino acids, an increase in Poly-P synthesis [39], and a greater entry of ions into the acidocalcisomes to restore cellular ionic strength. The driving force coupled with these ion fluxes is the H^+^ gradient, which is indirectly disrupted by cipargamin. In this way, cells maintained with cipargamin cannot respond to hyperosmotic stress (Figure 7). Taken together, these results indicate that cipargamin could be used in the future in association with current Chagas disease drug therapy. This fact is justified by the parasite’s ability to pass through the antiparasitic drugs effects. Combination therapy, in this case, is crucial to prevent drug adaptability.

## Figures and Tables

**Figure 1 membranes-15-00349-f001:**
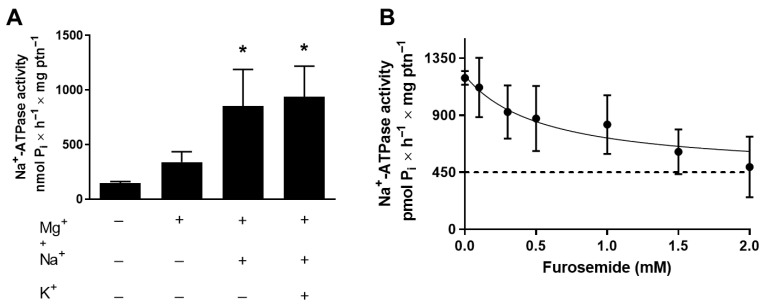
Na^+^-ATPase activity of *T. cruzi* is resistant to ouabain but sensitive to furosemide. (**A**) Total homogenate of *T. cruzi* epimastigotes was assayed using the methodology described in item 2.3, in the presence or absence of the following ions, as indicated in the abscissa: 10 mM MgCl_2_, 120 mM NaCl, or 30 mM KCl. (**B**) Dose–response curve of Na^+^-ATPase activity inhibition by furosemide. Total homogenates of *T. cruzi* epimastigotes were studied using the methodology described in item 2.3, in the presence of increasing doses of the inhibitor, as indicated on the abscissa. Values mean ± SE; *n* = 4. * *p* < 0.05. Statistical analysis was performed with one-way ANOVA.

**Figure 2 membranes-15-00349-f002:**
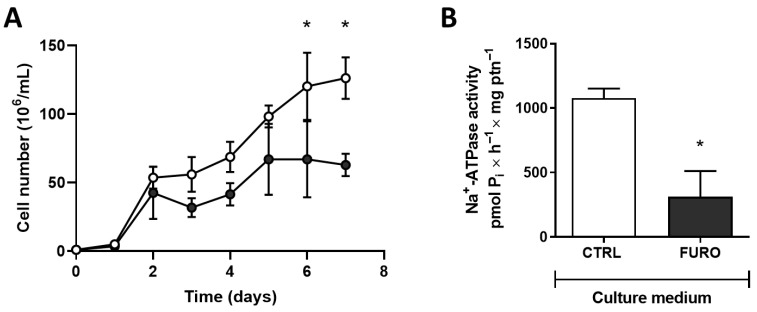
Inhibition of Na^+^-ATPase leads to a decrease in *T. cruzi* proliferation. (**A**) Proliferation of *T. cruzi* Dm28c epimastigotes maintained without (white circles) or with (dark circles) 2 mM of furosemide. Epimastigotes were collected in the stationary growth phase, washed twice with the PBS buffer, and subcultured in fresh medium for the times indicated on the abscissa. (**B**) Epimastigotes from 5-day proliferation culture, maintained in the absence (CTRL, white bar) or in the presence of 2 mM of furosemide (FURO, dark bar) were collected and washed, and the homogenate was assayed for Na^+^-ATPase activity, as described in item 2.3. Values are mean ± SE; *n* = 4. * *p* < 0.05 compared with CTRL. An unpaired Student’s *t*-test was used for statistical analysis.

**Figure 3 membranes-15-00349-f003:**
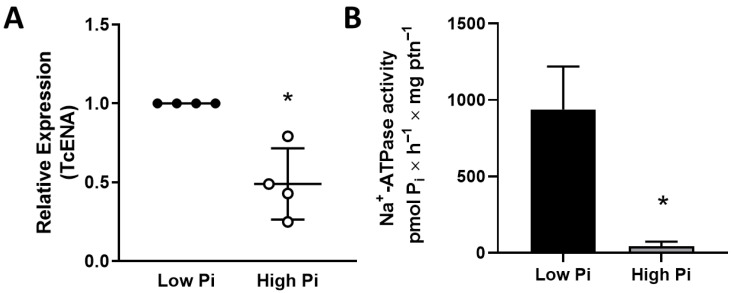
Relative expression of TcENA and Na^+^-ATPase activity of *T. cruzi* in high- or low-Pi medium. (**A**) Quantification of mRNA TcENA from *T. cruzi* epimastigotes. Quantitative PCR was performed using 100 ng of cDNA from each condition, as indicated on the abscissa (*n* = 4). (**B**) Total homogenate of *T. cruzi* epimastigotes was assayed, using the methodology described in item 2.3. Values are mean ± SE; *n* = 3. * *p* < 0.05 compared with Low Pi. Statistical analysis was performed with an unpaired Student’s *t*-test.

**Figure 4 membranes-15-00349-f004:**
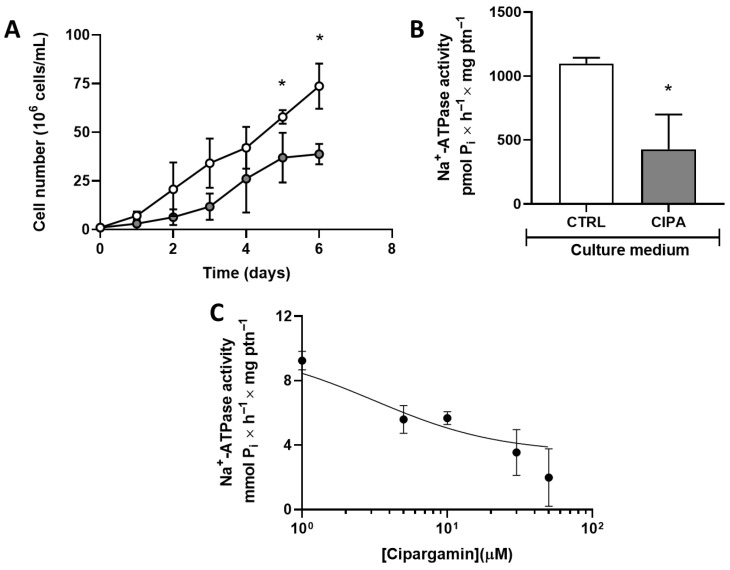
Cipargamin disrupts cell proliferation and dose-dependent inhibition of Na^+^-ATPase activity. (**A**) Proliferation of epimastigotes maintained in the absence (white circles) or presence of 50 nM (gray circles) cipargamin. Epimastigotes were collected in the stationary growth phase, washed twice in PBS buffer, and subcultured in fresh medium for the indicated times; values are mean SE; *n* = 6. (**B**) Na^+^-ATPase activity of parasites maintained in culture in the absence (CTRL, white bar) or presence (CIPA, gray bar) of 50 nM cipargamin. Total homogenates of *T. cruzi* epimastigotes were assayed using the methodology described in item 2.3. Values are mean SE; *n* = 6 (**C**) Dose–response curve of Na^+^-ATPase activity inhibition by cipargamin. Total homogenates of *T. cruzi* epimastigotes were used following the methodology described in item 2.3, in the presence of increasing cipargamin concentrations, as indicated on the abscissa. Values are mean ± SE; *n* = 3. * *p* < 0.05. An unpaired Student’s *t*-test was used for statistical analysis.

**Figure 5 membranes-15-00349-f005:**
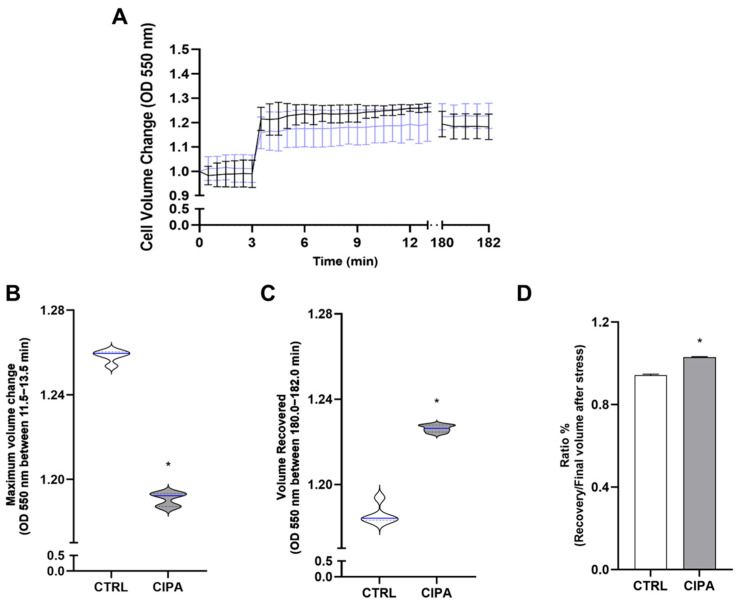
Cipargamin impairs parasite response to hyperosmotic stress and increases regulatory volume (RVI). (**A**) Epimastigotes were suspended in an isosmotic solution for 3 min, and hyperosmotic stress was induced for 10 min, resulting in a final osmolarity of 650 mOsm under constant ionic conditions. A light scattering technique was used to monitor changes in cell volume at 550 nm of control parasites (CTRL—black line) and parasites grown in the presence of 50 nM cipargamin (CIPA—blue line). Absorbance values were normalized by the first volume obtained (initial volume) and expressed as a percentage. (**B**) The maximum volume change reached under hyperosmotic conditions was observed by absorbances obtained in the last 2 min of reading (11.5 min to 13.5 min). (**C**) Absorbances recorded between 180 and 182 min of reading were observed in the final volume recovery. The first and third quartiles represent 25% and 75% of the cell volume values, respectively (black dashed lines). The interquartile range indicates the median of the cell volume values (blue line). (**D**) Ratio between absorbances observed at recovery volumes per absorbances observed at maximum volume change. Values close to 1 indicate no difference between these two states. Lower than 1 indicates high absorbance in the recovery volume. Values are mean ± SE; *n* = 3. * *p* < 0.05. The unpaired Student’s t-test was used for statistical analysis.

**Figure 6 membranes-15-00349-f006:**
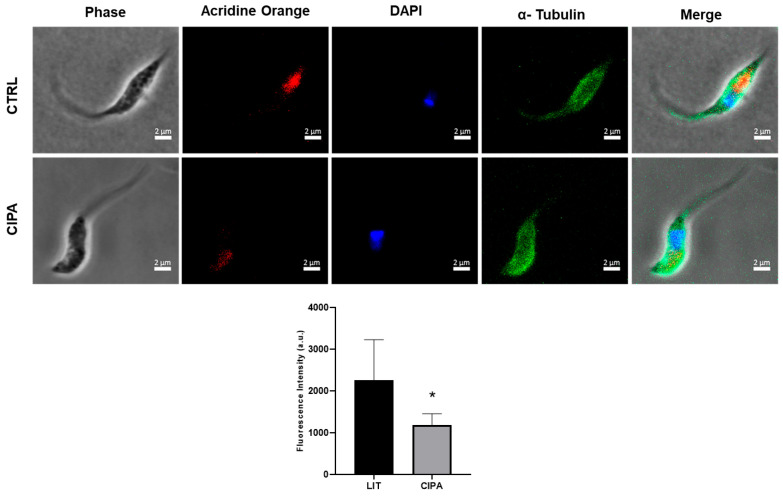
Immunofluorescence microscopy of *T. cruzi* epimastigotes grown in LIT medium with or without 50 µM cipargamin. Phase contrast. Acid compartments staining with acridine orange (red). Nucleic content staining with DAPI (blue). Control for preservation of cell membrane morphology staining with antibody for α-tubulin and the secondary anti-mouse-Alexa Fluor488^TM^ (green). Merge shows the overlay of the respective four signals. Percentage of parasites with acridine orange staining for 45 parasites obtained through three individual assays: *n* = 3. Magnification 100×. Scale bar 2 µm. * *p* < 0.05 compared with LIT using an unpaired Student’s *t*-test, statistical analysis was carried out.

**Figure 7 membranes-15-00349-f007:**
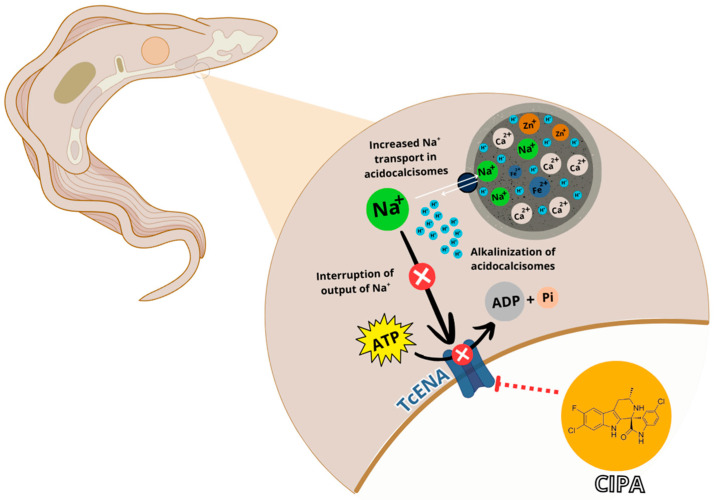
Proposed model for cipargamin action in *Trypanosoma cruzi*. Na^+^-ATPase (TcENA) inhibition leads to an alkalization of acidocalcisomes and interferes with the recovery volume increases (RVI) in *Trypanosoma cruzi*. Na^+^-ATPase inhibition by cipargamin blocks Na^+^ fluxes and harms RVI. Na^+^ is diverted to the acidocalcisome to avoid oxidative damage through Na^+^/H^+^ exchange. In this way, the Na^+^ increase follows H^+^ exit. This ion flux disrupts acid homeostasis and the ionic content of acidocalcisomes, which makes it impossible to recover lost cell volume in response to hyperosmotic stress. Red blunt arrow and ⊗ indicates inhibition by cipargamin (CIPA). Black arrows: mechanism of action of TcENA. Canva and NIH Bioart icons were used in the creation of this figure.

**Table 1 membranes-15-00349-t001:** Primer sequences for analyzed genes.

Primers Name	Forward Primer	Reverse Primer
TcENA	GCTCCTTTG CCGTCGGGG TC	GGCGAGAACACCCCAGTGCC
TcGAPDH	GCAGCTCCATCTACGACTCC	AGTATCCCCACTCGTTGTCG

## Data Availability

The original contributions presented in the study are included in the article. Further inquiries can be directed to the corresponding author.

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
