# Peer review of "Impacts of Cipargamin on Na+-ATPase and Osmoregulation of Trypanosoma cruzi"

_membranes, 2025, doi:10.3390/membranes15120349_

Round 1

Reviewer 1 Report

Comments and Suggestions for Authors

This study provides new insights into the role of ENA-type Na⁺-ATPases in parasitic protozoa and highlights their potential as selective therapeutic targets. The demonstration that cipargamin disrupts Na⁺ homeostasis in T. cruzi and impairs the parasite’s osmoregulatory capacity is both relevant and innovative. Since ENA- and ATP4-type ATPases are absent in mammalian cells but present in several protozoan parasites, the findings underscore the translational potential of targeting these proteins in the development of novel antiparasitic therapies, including future combination strategies for Chagas disease.

Overall, this manuscript addresses a relevant topic with potential significance in the field. However, several issues related to data clarity, formatting, consistency, and figure presentation need to be addressed before the work can be considered for publication.

See the Word document. 

Author Response

Reviewer #1

General comment.

This study provides new insights into the role of ENA-type Na+-ATPases in parasitic protozoa and highlights their potential as selective therapeutic targets. The demonstration that cipargamin disrupts Na+ homeostasis in T. cruzi and impairs the parasite´s osmoregulatory capacity is both relevant and innovative. Since ENA- and ATP4-type ATPases are absent in mammalian cells but present in several protozoan parasites, the findings underscore the translational potential of targeting these proteins in the development of novel antiparasitic therapies, including future combination strategies for Chagas disease.

Overall, this manuscript addresses a relevant topic with potential significance in the field. However, several issues related to data clarity, formatting, consistency, and figure presentation need to be addressed before the work can be considered for publication.

Response: We thank the reviewer for the nice comment.

Major comments.

Citation #1. Figure 1A: The figure legend and corresponding text should be rewritten for clarity. The concentrations of ions used are not specified in the legend or in the Methods section. In relation to the ions, in Figure 2, the concentration of Na+, K+ and Mg2+ used to measure ATPase activity should be specified.

Response: It is now described the ions concentration on figure legend 1A, and this is stated in the M&M section. It is now explained that after this result, the following experiments are performed in the absence of K+.

Citation #2: Figure 3: Clarify the definitions of “low Pi” and “high Pi”, and provide justification for the chosen concentrations.

Response: In RM, M&M section, sub-section 2.1., it was now specified the modified LIT low and high Pi medium

Citation #3: Figure 4A: The figure legend states that epimastigotes were maintained in the absence (white circles), or in the presence of 5 nM (light gray) or 50 nM (dark gray) cipargamin, but the figure only shows white and gray circles. This discrepancy should be resolved.

Response: We thank the reviewer for this comment. The figure legend was wrong; in RM we corrected this discrepancy.

Citation #4: Figure 5A: The figure should include a title and units (e.g., cell volume change (OD 550 nm)), similar to Figures 5B/C. Moreover, the x-axis currently lacks both a title and units.

Response: New figure 5A has title and units, similar to the figures 5B and C.

Minor comments.

Citation #5: Line 77: A closing bracket ] appears to be missing after reference 18.

Response: It is now corrected.

Citation #6: Line 81: There is a typographical error (“uhas”) that should be revised.

Response: The corrected word “has” was placed instead.

Citation #7: Line 105: In the Materials sections, some entries include both city and country, while others do not. The authors should adopt a single format and apply it consistently throughout.

Response: In RM, the companies are referred to without city or country.

Citation #8: Line 194: Include the error for the IC50 value.

Response: The value of IC50 for furosemide is now represented with mean ± error.

Citation #9: Line 197: The text provided does not accurately describe the corresponding figure. A revision is recommended to ensure clarity and accuracy.

Response: It was not clear if the text mentioned was referred to figure 1 or 2, because line 197 was in figure 1, not in text. Anyway, both texts referred to the mentioned figures were revised, in order to ensure clarity and accuracy. Moreover, legend figure 2B was rewrite for better accuracy as well.

Citation #10: Line 228: Please add a reference to support the sentence: “Then, cipargamin, a specific P-ATP4-type inhibitor [x].”

Response: New reference 22 was added to support the referred sentence.

Citation #11: Line 368: There is a strikethrough “s” in the word “arrows” that should be corrected.

Response: It was corrected in RM.

Citation #12: Line 378: The sentence beginning with Funding: Please add: This research was… is not properly integrated. “The use of ‘Please add” is awkward, consider rephrasing for clarity.

Response: We remove the sentence “Please add” in the indicated phrase.

Citation #13: Figures (general): The statistical reporting should include “±” between mean and SE, and a period should be placed after the sample size (e.g., n = 6).

Response: Now, all the legend figures present the statistical report with mean ± SE, with the period after the sample size.

Citation #14: Figure 4C: Add “Na+-ATPase activity” to the y-axis label, consistent with Figure 4B.

Response: New figure 4C has the title “Na+-ATPase activity” to the y-axis label, like the figures 4B.

Citation #15: Figure 5A: The error bars are difficult to visualize. The authors are encouraged to use consistent colors (e.g., blue and black) and to add the missing units on the x-axis.

Response: Error bars in new Figure 5A are now in blue and black, to better visualize the information.

Citation #16: Figures 5B and 5C: The data points within the diagrams are difficult to discern. The clarity of these figures should be improved.

Response: New Figures 5B and 5C are enhanced to better visualize the information within.

Citation #17: Figure 6: The percentage of parasites with Acridine Orange staining is reported for 45 parasites. Please clarify how many independent parasite cultures were analyzed to obtain there 45 parasites.

Response: The percentage of parasites stained with Acridine Orange was quantified for 45 parasites obtained through three individual assays, n = 3. This information is stated now on legend figure 6.

Citation #18: References: Reference 2 is the only one thar includes a DOI. The reference list should be revised to ensure consistent formatting across all entries.

Response: We removed DOI from the references, to present a consistent format throughout the manuscript.

Additional suggestions.

Citation #19: Since the manuscript proposes cipargamin as a candidate for combination therapy, the authors could test or discuss potential synergy or antagonism with standard Chagas disease drugs to enhance the translational relevance of the study.

Response: T. cruzi presents a growth plasticity and metabolic flexibility, what is determinant for parasite persistence through varied growth conditions or even under drug pressure (new reference 10.1016/j.mib.2021.07.017). This fact justified the parasite ability to pass through the anti-parasitic drugs effects. Combination therapy, in this case, is crucial to prevent drug adaptability. This is stated and discussed in RM, discussion section.

Citation #20. The authors could test or discuss the proposed biochemical responses, including polyphosphate synthesis and protein hydrolysis, and whether cipargamin blocks these adaptative responses.

Response: We are now discussing this topic in the discussion section in RM, as follows:

The ability of T. cruzi to withstand the vast array of environmental circumstances it experiences throughout its life cycle, including sharp variations in external osmolarity, is one unique aspect of its biology. As the insect's fed status varies, these osmotic fluctuations also occur in the triatomine vector's gut [new reference doi: 10.1016/S0022-1910(00)00170-0]. Acidocalcisomes and poly P hydrolysis have been shown to play a part in the regulatory volume drop that occurs in many trypanosomatids in response to hypoosmotic stress. So, alterations in poly P concentration could result in osmotic response defects [new reference doi:10.1042/BJ20070612].

Citation #21: To further strengthen mechanistic insight, experiments that modulate Na+/H+ exchangers or the proton gradient, either pharmacologically or genetically, could clarify the link between Na+ entry, H+ gradients, and osmoregulatory failure.

Response: It is already known that in T. cruzi, activation of Na+/H+ exchanger in acidocalcisomes is favored by a gradient of Na+ between the extracellular media and the organellar lumen and induces Ca2+ release from acidic vacuoles. Ca2+ release from acidocalcisome is crucial to respond to high osmolarity [new reference doi: 10.1016/j.abb.2012.07.014]. So, we discuss this issue better in the discussion section, highlighting the need for further studies aiming at the mechanistic response.

Reviewer 2 Report

Comments and Suggestions for Authors

In this manuscript, Dick et al presented the effects of cipargamin on T. Cruzi Na-ATPase. The study is well-designed, and the results are convicing. I have a few comments on the presentation of data, and rationale here for authors to consider. 

(A) From the references authors cited (refs. 22, 23, 24), the direct effects of cipargamin on TcENA is not clearly established, and the data presented here may not validate a direct connection between cipargamin and TcENA. I understand that it may be out of the scope of this paper to prove a direct connection, the authors may need to acknoeledge and discuss it in discussion section.

(B) Comments on specific data and data presentation: 

(1) Fig. 2A, the authors would need to clarify why there is a dip in cell density between days 2-4 which was not observed in Fig. 4A.

(2) Fig 3A, the data spread in High Pi column seems to be too big to warrant a statistical significance.

(3) Fig. 4A, the hollow points here means a lower dose of drug whereas in Fig. 2A means no drug present, I would suggest against this inconsistent data presentation.

(4) Fig. 4C, it would be more clear if X-axis is presented in log scale. 

(5) The data presented in Fig. 6 seems disconnected with other content of this psper. Please provide some more background before paragraph starting from Line 277, and perhaps include more in dicsussion. 

Author Response

Reviewer #2

General comment.

In this manuscript, Dick et al presented the effects of cipargamin on T. Cruzi Na-ATPase. The study is well-designed, and the results are convincing. I have a few comments on the presentation of data, and rationale here for authors to consider. 

Response: We thank the reviewer for the nice comment. We addressed the comments as follows.

Major comments.

Citation #1. (A) From the references authors cited (refs. 22, 23, 24), the direct effects of cipargamin on TcENA is not clearly established, and the data presented here may not validate a direct connection between cipargamin and TcENA. I understand that it may be out of the scope of this paper to prove a direct connection, the authors may need to acknowledge and discuss it in discussion section.

Response: The specific effect of cipargamin on PfATP4 protein is stated on Jimenez-Diaz et al., 2014 (new reference 22). And in silico analysis indicates that TcENA has close structural similarity with PfATP4, and both proteins are close through phylogenetic analysis (new reference 11). This is stated and discussed in the discussion section in RM.

Citation #2. (B) Comments on specific data and data presentation: 

(1) Fig. 2A, the authors would need to clarify why there is a dip in cell density between days 2-4 which was not observed in Fig. 4A.

Response: We thank the reviewer for this comment. We revised the data, and presented the new Figure 2A, with no dip on control conditions (white circles). However, both Figure 2A and 4A are related to different proliferation assays, with individual controls groups in each. Although we expect a similar response, it cannot be the same curve. Moreover, in Figure 2A, the observed dip on 3-days proliferation on cells maintained in the presence of furosemide, but it was not significantly through t-student test.

Citation #3. (2) Fig 3A, the data spread in High Pi column seems to be too big to warrant a statistical significance.

Response: We thank the reviewer for this comment. We added one more n in this experiment, with now a P=0.004 with an unpaired student t-student.

Citation #4. (3) Fig. 4A, the hollow points here means a lower dose of drug whereas in Fig. 2A means no drug present, I would suggest against this inconsistent data presentation.

Response: In both Figures 2A and 4A, white symbols mean control conditions, i.e., in the absence of drugs.

Citation #5. (4) Fig. 4C, it would be more clear if X-axis is presented in log scale. 

Response: New Figure 4C, x-axis is now presented in log scale.

Citation #6. (5) The data presented in Fig. 6 seems disconnected with other content of this paper. Please provide some more background before paragraph starting from Line 277, and perhaps include more in discussion. 

Response: This information required is essential to a better understanding of the following data. For that, we thank the reviewer. A more detailed background is added to results section, with some new references (doi:10.1042/bj3040227, doi:10.1128/spectrum.01128-24; doi:10.1074/jbc.M204744200; doi:10.1128/mmbr.00042-23). In discussion section, we also discussed better the relationship between Na+-ATPase, Na+ gradient and osmotic effect.

Round 2

Reviewer 1 Report

Comments and Suggestions for Authors

The revisions and responses to the concerns have been mostly addressed. However, the similarity index remains relatively high at 27%. I recommend that the authors further revise the manuscript to reduce redundant or overlapping text and ensure that any remaining similarity is limited to essential methodological or background sections.

Author Response

Response to Reviewers

Manuscript membranes-3913079

The manuscript entitled “Impacts of cipargamin on Na+-ATPase and osmoregulation of Trypanosoma cruzi” has been thoroughly revised following the recommendation by Reviewer 1.

The modified parts are highlighted in clear blue throughout the R2 version. The yellow markings indicating modifications made in the revised R1 version are preserved in this R2 version.

We appreciate the Reviewer’s revision. The corrections are detailed as follows.

Reviewer 1.

Comment.

Citation: The revisions and responses to the concerns have been mostly addressed. However, the similarity index remains relatively high at 27%. I recommend that the authors further revise the manuscript to reduce redundant or overlapping text and ensure that any remaining similarity is limited to essential methodological or background sections.

Answer: We addressed the comment in four parts.

  1. Upon reviewing the phrases and words used to compute the similarity index, we noted that most do not indicate redundancy or overlap with results of our previous publications or those of others. These markings range from parasite names to the mention of t-tests, and even to the section crediting the authors. Please, revise this issue.
  2. However, we made a first effort in the Abstract to address the Reviewer's recommendation. The Abstract has been rewritten, avoiding most of the words marked in red after R1 (page 1, lines 16 to 30; R2).
  3. Introduction. Even though our Introduction is mostly a background section, we. revised this section, replacing – whenever possible – words that were considered redundant during the first revision. The same is true for the Materials and Methods; this is a section where there are few marked words. See: page 1, line 34 to page 5, line 190; R2)
  4. Results. Again, we revised this section, replacing – whenever possible – words that were considered redundant during the first revision (page 5, lines 192 to page 9, lines 303; R2). This is also true for the Discussion (page 9; line 305 to page 11, line 372; R2).

Round 3

Reviewer 1 Report

Comments and Suggestions for Authors

The revisions and responses to the concerns have been addressed.

Author Response

Reviewer 1.

Comment.

Citation: According to the text similarity check, there are several part of hight text similarities and those need to be revised before acceptance of this paper.

Answer: We revised the entire manuscript, replacing, whenever possible, words considered redundant during the previous revision. However, as observed in these new similarity reports sent to us, although there are fewer similar words, the index is higher now (37%) than it was in R2 (27%), which made our work a lot more difficult, as we do not know if the new modifications could increase the similarity index. We hope that, with all new modifications, the R3 version will be considered acceptable for publication in Membranes.